# Calculation of the Vapour Pressure of Organic Molecules by Means of a Group-Additivity Method and Their Resultant Gibbs Free Energy and Entropy of Vaporization at 298.15 K

**DOI:** 10.3390/molecules26041045

**Published:** 2021-02-17

**Authors:** Rudolf Naef, William E. Acree

**Affiliations:** 1Department of Chemistry, University of Basel, 4003 Basel, Switzerland; 2Department of Chemistry, University of North Texas, Denton, TX 76203, USA; acree@unt.edu

**Keywords:** group-additivity method, vapour pressure, Gibbs free energy of vaporization, entropy of vaporization

## Abstract

The calculation of the vapour pressure of organic molecules at 298.15 K is presented using a commonly applicable computer algorithm based on the group-additivity method. The basic principle of this method rests on the complete breakdown of the molecules into their constituting atoms, further characterized by their immediate neighbour atoms. The group contributions are calculated by means of a fast Gauss–Seidel fitting algorithm using the experimental data of 2036 molecules from literature. A ten-fold cross-validation procedure has been carried out to test the applicability of this method, which confirmed excellent quality for the prediction of the vapour pressure, expressed in log(pa), with a cross-validated correlation coefficient Q^2^ of 0.9938 and a standard deviation σ of 0.26. Based on these data, the molecules’ standard Gibbs free energy ΔG°_vap_ has been calculated. Furthermore, using their enthalpies of vaporization, predicted by an analogous group-additivity approach published earlier, the standard entropy of vaporization ΔS°_vap_ has been determined and compared with experimental data of 1129 molecules, exhibiting excellent conformance with a correlation coefficient R^2^ of 0.9598, a standard error σ of 8.14 J/mol/K and a medium absolute deviation of 4.68%.

## 1. Introduction

In recent years, knowledge of the vapour pressure of organic molecules has gained increasing interest in view of the environmental, in particular radiation absorption, effects in the context of global warming, but also in view of their toxicology [1,2], as well as their quality as refrigerants [3]. At the same time, new and highly sophisticated experimental methods, e.g., using a Knudsen effusion apparatus, coupled with a quartz crystal microbalance [4], have been developed for the measurement of molecules exhibiting extremely low vapour pressures. In most cases, these measurements involved the temperature dependence of the vapour pressure over a certain temperature range, the corresponding sequence then being approximated by one of various non-linear functions, mostly by the Antoine equation [5]. In order to enable a comparison of the vapour pressures between molecules at identical conditions, the non-linear functions have been used to interpolate the vapour pressures to a standard temperature, usually 298.15 K. This interpolation method produces reliable results on condition that the experimental temperature range encompassed the standard temperature. However, the comprehensive handbook of Mackay et al. [6], collecting the experimental data from various authors for more than 1000 compounds, revealed that in many cases the standard vapour-pressure data varied by a wide range, depending on the experimental methods. Therefore, many attempts, critically reviewed by O’Meara et al. [7] and Dearden [8], have been undertaken to calculate the vapour pressure based on quantitative structure-property relationships. Some of these attempts require the knowledge of an experimentally determined descriptor, which limits the scope of applicability for vapour-pressure prediction. For example, A. Vetere [9] suggested a non-linear equation relating the vapour pressure to the temperature, based on the reduced temperature T_f_ (i.e., the critical temperature), which was tested successfully on less than 50 liquids. A neural network approach was presented by R. Kühne et al. [10], whereby the network was trained by 1200 and tested by 638 hydrocarbons and halogenated hydrocarbons, requiring—among further molecular structure data—the melting point of the compounds, achieving an overall error of 0.08 and 0.13 log(Pa) for the training and test set. The authors stressed that neural networks cannot extrapolate reliably outside the given descriptor and target values of the training set. A group-contribution method with the inclusion of group interactions was presented by B. Moller et al. [11], which produced the group parameters, but required the knowledge of one experimental vapour pressure point. The relative error for a training set of 2332 compounds was given as 5%. Several other prediction methods are based on a set of purely structural descriptors and/or on atom groups of the molecules. For the calculation the vapour pressure of a large scope of molecules, these latter methods are entirely dependent on the number and structural variability of molecules with known vapour pressure. In 1994, Ch.-H. Tu [12] presented a group-contribution method based on 5359 experimental vapour pressure data of 342 compounds over a varying temperature range between 90 and 643K, whereby each atom group was defined by four constants, enabling the prediction at various temperatures using a second order equation derived from the Clausius–Clapeyron equation. The medium absolute percentage deviation between experimental and predicted values for 336 compounds was reported as 5%. The reliability of these predictions, however, is said to be limited to molecules carrying at most one functional group. A neural network model was used by E. S. Goll and P. C. Jurs [13] for the vapour-pressure prediction of hydrocarbons and halohydrocarbons, having been trained by the experimental vapour pressures as log(VP) at 25 °C of 352 compounds. The molecules were presented to the neural network by topological, geometric, electronic and hybrid descriptors. The root-mean-square (rms) deviations for the training, cross-validation and prediction sets were given as 0.163, 0.163 and 0.209 log units, respectively. An analogous model, this time based on 420 diverse molecules, was presented by H. E. McClelland and P. C. Jurs [14], yielded an rms error of 0.33 log units. Cohesive energies and solubility parameters derived from molecular dynamics simulations based on forcefield calculations of 22 molecules have been used by P. K. C. Paul [15]. He demonstrated that a single cross term consisting of either the molecular volume or molecular weight and the square of the compound’s solubility parameter—which latter is the square root of the cohesive energy—determined to more than 90% the equation for the vapour pressure, expressed as log(VP). The Abraham descriptors method has been used for the vapour-pressure prediction as log(VP) of liquid and solid organic and organometallic molecules by M. H. Abraham and W. E. Acree [16], the six descriptors being E the excess molar refraction, S the solute polarity/polarizability, A the solute H-bond acidity, B the solute H-bond basicity, V the McGowan’s characteristic molecular volume and L the logarithm of the hexadecane partition coefficient at 298.15 K, all of which, except for L, either being available from commercial databases for more than 8000 compounds or obtainable by calculation procedures. The best standard deviation value has been calculated to 0.28 units for 1016 compounds. These few examples (except for the last one) demonstrate that the studies on the prediction of the vapour pressure published up to now usually either deal with a specific set or a limited number of molecules forestalling an extension beyond them.

The present paper provides a way to predict the vapour pressure at 298.15 K of a very large scope of organic molecules, applying the same basic computer algorithm based on the atom-group additivity method outlined in [17], which has already proven its versatility in the reliable prediction of the 16 molecular descriptors enthalpy of combustion, formation, vaporization, sublimation and solvation, entropy of fusion, logP_o/w_, logS, logγ_inf_, refractivity, polarizability, toxicity (against the protozoan *Tetrahymena pyriformis*), viscosity and surface tension of liquids, and heat capacity of solids and liquids [17,18,19,20,21], and which only required a few further peripheral control lines of code to meet the present purpose. In the next section, a brief outline of the calculation procedure is given. In addition, by the inclusion of the experimental and calculated enthalpy of vaporization of the molecules under consideration (their calculated enthalpy having been received by the same method as the present one but published earlier [18]), their experimental and predicted entropies of vaporization have been made accessible and will thus be compared.

## 2. Method

The present study rests on a regularly updated object-oriented knowledge database of currently 32,697 compounds encompassing the entire spectrum of organic molecules, including pharmaceuticals, herbicides, pesticides, fungicides, textile dyes, ionic liquids, liquid crystals, metal-organics, lab intermediates, and more, each of them stored in a separate datafile containing the 3D-geometry-optimized structure and—as far as available—their experimental and routinely calculated descriptors, including their vapour pressures. The latter is defined in this work as the logarithm to the basis 10 in Pascal, termed as logVP.

The atom-group additivity principle and its translation into a computer algorithm for the calculation of their parameters has been outlined in detail in [17]. Accordingly, the definitions and naming of the atom and special groups are identical to the ones given in Table 1 and Table 2 of [17]. The first preliminary logVP calculations, however, with tentative replacement of certain atom groups by more detailed ones and addition or omission of certain special groups, revealed a significant improvement of the statistical data upon the addition of the groups explained in Table 1.

The separation of the hydroxy group on saturated carbon into primary, secondary and tertiary OH groups as defined in Table 1 (henceforth called “saturated hydroxy group”) has successfully been introduced into the present atom-group additivity approach for the calculation of the heat capacities of molecules [21]. This modification required an additional procedure in the group-additivity algorithm described in [17]. In contrast to this, the group definition of hydroxy groups attached to unsaturated carbon atoms remained unaltered. The tentative first calculations also confirmed an assumption which had already been proven in the calculation of the surface tension of liquids [20]: additional saturated hydroxy groups in a molecule exhibit more than just a linearly additive effect on the descriptor. This non-linear effect has been considered by the special group “(COH)n” for molecules carrying more than 1 saturated hydroxy group (the term “Neighbours” at the header of the second column does not apply to the special groups). Similarly, the first attempts for the prediction of the logVP for dicarboxylic acids indicated that a second carboxylic acid function also showed a nonlinearly increasing impact on the result. Therefore, the special group “(COOH)n” was added for compounds with n>1 to take account of this apparent nonlinearity. A further strong deviation between predicted and experimental data, observed with compounds containing cyclic saturated segments, was remedied by adding a correction value for each single bond that is a part of the ring moiety, defined by the special group “Endocyclic bonds”, yielding a drastic improvement of the prediction statistics data (summed up at the bottom of Table 3). This special group has already found successful use in the prediction of the entropy of fusion [18] and the heat capacities [21] of molecules. Despite this additional special group, the predicted vapour pressures for various bicyclic compounds such as adamantane or camphor have shown to be systematically much lower than their experimental data. Therefore, the special group “Bridgehead atoms” has been introduced. Further details about these special groups will be discussed in the results section.

The calculation of the parameter values of the atom and special groups of Table 3 is carried out in a step-by-step process as explained in [17]: in a first step, a temporary list of compounds for which the experimental vapour pressure is known, is extracted from the database. In a second step, for each of the “backbone” atoms (i.e., atoms bound to at least two other direct neighbour atoms) in the molecules the atom type and its neighbourhood is defined by two character strings according to the rules defined in [17], corresponding to the atom type and neighbours terms listed in Table 3, and then its occurrence in the molecule is counted. Thirdly, an M × (N + 1) matrix is generated where M is the number of molecules and N + 1 the molecules’ number of atom and special groups plus their experimental value and where each matrix element (i,j) receives the number of occurrences of the jth atomic or special group in the ith molecule. In the final steps, normalization of this matrix into an Ax = B matrix and its subsequent balancing using a fast Gauss–Seidel calculus, as e.g., described by E. Hardtwig [22], yield the group contributions, which are then stored in Table 3.

Following the philosophy of the group-additivity approach, these group contributions can now be used to calculate the descriptor, in this case the vapour pressure as logVP at 298.15 K, by simply summing up the contributions for each of the molecule’s atom and special group, according to Equation (1), wherein a_i_ and b_j_ are the respective atom and special group contributions, A_i_ is the number of occurrences of the ith atom group, B_j_ is the number of occurrences of the jth special group and C is a constant. However, an important restriction has to be observed when using Table 3 in connection with Equation (1): the group contributions are only reliable enough for use—i.e., “valid”—if they are supported by at least three independent molecules, i.e., if the number in the rightmost column of Table 3 exceeds 2.
(1)$$logVP=\underset{i}{\sum }a_{i}*A_{i}+\underset{j}{\sum }b_{j}*B_{j}+C$$

The plausibility of the descriptor results is immediately tested in the present method by means of a 10-fold cross-validation algorithm wherein in each of the 10 recalculations another 10% of the complete set of compounds is used as a test set, ensuring that each compound has been entered alternatively as a training and a test sample. The respective statistics data of the training and accumulated test calculations are finally collected at the bottom of Table 3. Due to the restriction mentioned above, the number of molecules for the evaluation of the training correlation coefficient, average and standard deviations (lines B, C and D) and for the corresponding test data from cross-validation (lines F, G and H) are smaller than the number of compounds shown on line A, upon which the complete list of atom-group parameters is based. The number of “valid” groups (line A) is significantly lower than the total of atom and special groups listed in Table 3, leaving a substantial number of “invalid” groups. Although not applicable for vapour-pressure predictions at present, they have deliberately been left in Table 3 for future use in this continuous project (and may motivate interested scientists to focus on measuring the vapour pressure of molecules carrying the under-represented atom groups). At present, the elements list for vapour-pressure predictions is limited to H, B, C, N, O, P, S, Si, and/or halogen.

A simple example may help to understand the application of the data of Table 3 in Equation (1): 2-methylcyclohexanol consists of the atom and special groups listed in Table 2. Accordingly, the sum of all the contributions is 2.08. The experimental logVP was published in [16] as 2.216.

## 3. Sources of Vapor-Pressure Data

An overview of the literature concerning the vapour-pressure data of molecules revealed that generally their measurements were either carried out at 298.15 K or over a certain temperature range encompassing this standard temperature. In the latter case, the authors mostly provided a set of constants to be used in a non-linear equation, usually the Antoine-equation [6], allowing to calculate the vapour pressure at the standard temperature by interpolation. However, in several publications the authors chose a temperature range which remained above this standard, particularly with compounds having a high melting point. On condition that the lowest experimental temperature was not higher than 5 K above the standard, the extrapolated vapour-pressure data have been included in the present study, well aware of the uncertainty of these values. Some authors overcame the problem of the high melting point by supercooling the melt prior to measuring the vapour pressure, examples of which are cited in [6]. While past publications usually expressed the vapour pressure in older units such as Torr, mm(Hg) or atm, newer ones generally used Pa, kPa, MPa or mPa, often converted to their natural logarithm. In the present paper, these various units have been translated throughout to the logarithm to the basis 10 of the unit Pa, expressed as logVP.

Several comprehensive papers provided the majority of the vapour-pressure data: D. Mackay’s Handbook [6] is a compilation of over 1000 compounds, collecting—among several further physico-chemical data—the results of the vapour-pressure measurements of various authors for each compound, revealing the general extent of the experimental uncertainty, depending on the method of measurement. Another rich source was the comprehensive collection of M. H. Abraham and W. E. Acree Jr [16], contributing the vapour-pressure data of additional more than 1000 compounds. Further collective sources have been used to complement—and compare—the experimental data [8,14,15,23,24,25,26,27,28,29,30]. In addition, vapour-pressure data have been published specifically for various saturated and unsaturated hydrocarbons [31,32,33,34,35,36,37,38,39,40,41,42,43,44,45,46,47,48,49,50,51,52], alcohols [53,54,55,56,57,58], phenols [59,60], alkyl- and arylethers [61,62,63,64], acetals [65,66], carboxylic acids [67,68,69,70,71], carboxylic halides [72], carboxylic esters and lactones [73,74,75,76,77,78,79,80,81,82,83,84,85,86], carbonates [87,88,89,90,91], ketones [92,93,94], peroxides [95,96], amines [97,98,99,100,101,102,103,104,105,106], amides [107,108,109,110], azides [111,112], hydrazines [113,114,115], isocyanates and isothiocyanates [116,117,118,119], nitriles [120,121], nitro-substituted compounds [122,123,124,125,126], nitrites [127], nitrates [128], nitrosamines [129], ureas [130,131], alkyl- and arylsulfides [132], sulfoxides [133,134,135,136], thiophenes [137,138,139], phosphines and phosphoranes [140,141,142,143,144], phosphonates, phosphates and thiophosphates [145,146,147,148,149,150,151,152], boranes and borates [153,154,155,156,157,158,159,160], silanes, siloxanes and silthianes [161,162,163,164,165,166,167,168,169,170,171,172,173,174,175] and hetarenes [176,177,178,179]. A particularly large number of publications studied the vapour-pressure data of halogen-substituted compounds, which in many cases belonged to one of the aforementioned groups, hereinafter subdivided in haloalkanes [180,181,182,183,184,185,186,187,188,189,190], haloalkenes [191,192,193,194,195,196,197,198,199,200,201,202], haloaromatics [203,204,205,206,207,208,209,210,211], haloalkyl-amines, -aminoxides and -alkoxyamines [212,213,214,215,216,217], haloalkylethers [218,219,220,221,222,223], haloarylethers [224], haloalkylsulfides, -sulfoxides, -sulfones and -sulfates [225,226,227,228,229,230,231], haloalkylsilanes [232,233], haloalkylcarboxylic acids, -esters, and peresters [234,235]. Finally, a number of authors published the vapour pressures of some individual compounds, as there were the terpenes and their derivatives carvone, 2-hydroxy-3-pinanone, iso-pinocampheol, myrtanol, pinocarveol, eugenol, camphor, menthone, damascenone and (-)-methyl jasmonate [236,237,238], several hydroxycarbonyl- and formyl-derivatives of naphthalene, fluorene, anthracene and pyrene with exceptionally low vapour pressure [239], 2-aminoethanol and its *N*-methyl derivatives [240], 2-dialkylaminoethanethiols [241], derivatives of 2- and 3-amino-1-propanols [242], phthalan [243], the aroma compounds d-linalool, 2-nonanone, d-limonene and isoamyl acetate [244], hydroxyacetaldehyde and hydroxyacetone [245], l-deprenyl, benzphetamine and alverine [246], 2-adamantanone and 1-acetyl-adamantane [247], fenpropidin and phencyclidine [248], ambroxide and galaxolide [249], the lignin fragments *trans*-anethole, estragole, eugenol as well as hydroxy- and methoxy-substituted benzaldehydes [250,251,252,253], benzocaine [254] and bicifadine [255].

## 4. Results

### 4.1. Vapour Pressure

The contributions of the atom groups in Table 3 for the prediction of the vapour pressure as logVP in Pascal are the final result of a series of direct and cross-validation calculations according to the method outlined in [17], whereby in preceding step-by-step calculations any outliers, defined by the deviation of their experimental from their predicted value exceeding three times the current cross-validated standard error Q^2^, have been removed from further calculations. At the end, they made up ca. 11% of the total number of molecules for which experimental vapour pressures were given, which have been collected in an outliers list accessible in the Appendix A. The statistical data are collected at the bottom of Table 3 in rows A to H. As is shown in row A, of the 314 atom and special groups required to cover all remaining 2036 compounds, only 171 groups have been found to be valid for predictions. Accordingly, the number of compounds, for which a prediction was possible, has been reduced to 1908 in the complete training set and to 1842 in the cross-validation test sets. The high compliance of the direct and the cross-validation correlation coefficients R^2^ and Q^2^ (lines B and F), also evident in the low scatter of both the respective data points about the correlation line in the corresponding diagram (Figure 1) and confirmed by the narrow symmetrical Gaussian bell curve of the histogram (Figure 2), confirms the excellent reliability of the present atom-groups approach for vapour-pressure predictions. Its low cross-validation standard deviation of only 0.26 units compares very favourably with the best values of other prediction methods such as that of McClelland and Jurs [14] or that of Abraham and Acree [16], although it is based on a much larger number of molecular structures. The large range of experimental vapour pressures of between ca. 10^7^ Pa for tetrafluoroethylene and 10^−19^ Pa for hexapentacontane and the broad structural variety of molecules, upon which these calculations are based, enabled the trustworthy prediction of the vapour pressure of more than 57% of the compounds listed in the database which can well be considered as representative for the entire realm of chemical structures. A list of the molecules with their experimental and predicted vapour pressure data is accessible in the Appendix A.

A few observations concerning certain atom and special groups are worth being highlighted: separation of the hydroxy group at a saturated carbon atom into primary, secondary and tertiary OH groups (group numbers 209–211 in Table 3) led to an appreciable improvement of the statistics data. A comparison of their individual parameter values reveals that primary hydroxy groups generally cause noticeably lower vapour pressures than secondary or tertiary analogues. This pattern matches with the observation made in their influence on the heat capacities of molecules [21,256]. It may be explained by an effect which was discussed in studies by Huelsekopf and Ludwig [257] which, based on the quantum cluster equilibrium theory (QCE), demonstrated that primary alcohols principally form cyclic tetramers and pentamers in the liquid phase, whereas tertiary alcohols only form mono- and dimers. (Secondary alcohols have not been considered.) This clustering of primary alcohols could also prevail in the gas phase, consequently leading to a lower saturated vapour pressure.

Another peculiarity was found with di- and trihydroxyalkyl compounds in that the negative impact of any additional OH group on the vapour pressure was clearly larger than just cumulative. This effect was taken account of by the special group “(COH)n” which was invoked for n > 1. Similarly, and even more drastically, the second carboxylic function in dicarboxylic acids lowered the vapour pressure in more than a cumulative way, which required the additional special group “(COOH)n”, again called up for n > 1. Both these nonlinearities have already been reported by Compernolle et al. [258] and have been considered by additional parameters in their development of EVAPORATION, a group-additivity model for vapour-pressure prediction especially designed for secondary organic aerosols (SOA) comprising alkyl and alkenyl compounds optionally carrying various functional groups but ignoring aromatic systems. They did not provide a reason for the nonlinearities of the additional OH and COOH groups on the vapour pressure; the seemingly obvious argument pointing to intermolecular hydrogen-bridge effects is questionable in view of their observation of a similar nonlinearity with polynitrates. It should be mentioned however, that the present method did not require an additional parameter for the vapour-pressure prediction of polynitrates.

Linear and cyclic, unbranched alkanes having the same number of carbon atoms exhibit comparable experimental vapour pressures (in logVP), e.g., butane vs. cyclobutane: 5.38 vs. 5.195, pentane vs. cyclopentane: 4.84 vs. 4.62, hexane vs. cyclohexane: 4.3 vs. 4.11, heptane vs. cycloheptane: 3.78 vs. 3.45, or octane vs. cyclooctane: 3.27 vs. 2.876 (cited from [25]). For the prediction of logVP of the linear alkanes, the present method simply sums up the contributions of the n-2 methylene groups (i.e., n-2 x -0.47) and adds twice the contribution of the end methyl groups (i.e., 2 x +0.6) to the constant (4.71). In cyclic alkanes however, the two end methyl groups with their large positive contributions are replaced by two methylene groups contributing with large negative values. Therefore, in order to still achieve the goal of vapour pressures comparable to their linear counterparts, the methyl-methylene replacement effect had to be compensated. This was achieved by the introduction of special group “Endocyclic bonds”, whereby its parameter value of +0.31 represents the additional contribution of each single bond of the cyclic moieties of the molecule. For 3-, 4- and 5-membered saturated rings the special groups “Angle60”, “Angle90” and “Angle102” (successfully used for the calculation of the heats of combustion [17]) have been added to Table 3 to take account of their further increasing effect on the vapour pressure. Yet, it turned out that for bicyclic molecules, e.g., camphor or adamantane and its derivatives, the combined contributions of these special groups are still too small to compensate for the even larger negative atom-group contributions of the three- or four-bonded atoms at their bridge heads, (defined by e.g., “C sp3 / HC3”: −1.28, “C sp3 / C4”: −2.19, or “C sp3 / C3O”: −3.46). As a consequence, the special group “Bridgehead atoms” had to be introduced, successfully lifting this deficiency by the additional parameter value of +0.23 units for each bridgehead atom.

### 4.2. Gibbs Free Energy of Vaporization

Some authors [28,29] derived the experimental vapour pressure of molecules from the experimental data of their enthalpy and entropy of vaporization or sublimation at standard conditions, applying Equation (2), wherein ΔG° is the Gibbs free energy, ΔH° the enthalpy and ΔS° the entropy of vaporization/sublimation, and Θ the reference temperature of 298.15 K. By insertion of ΔG° into the integrated Clausius–Clapeyron Equation (3), wherein p° is the standard pressure of 101,325 Pascal and R the gas constant, and assuming ideal gas-phase conditions and neglecting the volume of the condensed phases, they received the vapour pressure p at 298.15 K in Pascal.
ΔG°_vap,sub_(Θ) = ΔH°_vap,sub_(Θ) − ΘΔS°_vap,sub_(Θ)(2)
p(Θ) = p°exp[−ΔG°_vap,sub_(Θ)/(RΘ)](3)

In a logical inversion of the mathematical approach, the vapour pressures, calculated by the present group-additivity method, have been used to predict the Gibbs free energy in kJ/mol by simply using the rearranged form of Equation (3), i.e., Equation (4), and focusing on vaporization.
ΔG°_vap_(Θ) = −RΘln(p(Θ)/p°) = −RΘln(10^logVP^/101325)(4)

Applying Equation (4) on the experimental and predicted vapour pressures yielded the correlation diagram of the Gibbs free energies in Figure 3. Evidently, since both ΔG°_vap_ are simple translations according to Equation (4), their correlation coefficient is identical with that in Figure 1 for the vapour pressures. A list of the molecules with their experimental and predicted free energies is available in the Appendix A.

### 4.3. Standard Entropy of Vaporization

The standard entropy of vaporization ΔS°_vap_(Θ) of a molecule can be calculated from the Gibbs free energy ΔG°_vap_(Θ) using Equation (5), which is the rearranged form of Equation (2), provided that the standard enthalpy of vaporization ΔH°_vap_(Θ) is known. The present database of currently 32697 molecules has the advantage of encompassing—besides the experimental vapour pressures for 2036 samples—the experimental heat-of-vaporization data for 4029 compounds and, based on these, the predicted heats of vaporization for 24309 compounds, calculated by means of the same group-additivity method as the present one, described in [18]. Hence, it was exciting to compare the results of Equation (5), if in the first case both input data have been experimental values and in the second case both originate from predicted data.
ΔS°_vap_(Θ) = [ΔH°_vap_(Θ) − ΔG°_vap_(Θ)]/Θ(5)

Accordingly, the correlation diagram in Figure 4 compares the entropies of vaporization received from both the experimental enthalpies and energies of vaporization with those calculated from both the predicted enthalpies and energies, revealing a surprisingly small medium absolute percentage deviation (MAPD) of less than 5%. The corresponding histogram in Figure 5 confirms the narrow scatter about the correlation line. The limited number of only 1129 samples in this diagram is owed to the fact that only for these both experimental enthalpies and free energies (or more precisely: vapour pressures) have been available, whereas the large number of calculated enthalpies and free energies (>20800) in the database enabled the reliable prediction of the entropies of vaporization for 20232 (i.e., ca. 62%) compounds of the database. A list of the compounds with their experimental and predicted entropies of vaporization as well as their experimental and predicted enthalpies of vaporization have been added to the Appendix A.

## 5. Conclusions

The present paper is the result of an extension of a common group-additivity approach applied in an ongoing software project enabling the direct and indirect calculation of 16 physical, thermodynamic, solubility-, optics-, charge- and environment-related descriptors, which led to a series of earlier publications [17,18,19,20,21]. The present project extension, enabling the trustworthy prediction of the vapour pressure and subsequent Gibbs free energy of vaporization at 295.15 K of molecules, also immediately allowed the reliable calculation of the molecules’ standard entropy of vaporization due to the project’s direct access to their predicted heat of vaporization as outlined in [18]. The big advantage of the present group-additivity approach, encoded in the common computer algorithm outlined in [17], not only rests on its simple extensibility by the addition of just a few further lines of control code to fulfil the present task, but also in its simple applicability, basically even allowing accurate prediction of any of the mentioned molecular descriptors by means of paper and pencil and usage of their corresponding group-parameters table. A further advantage is the easy extensibility of the group-parameters lists (if required) to take account of molecules with known descriptor values newly added to the database, usually followed by a recalculation of the group parameters. The disadvantage of the large number of group parameters due to the radical breakdown of the molecules and its subsequent limitation of the calculations to molecules only, for which all the group parameters are found in the respective tables, is well compensated by the high accuracy of the predicted values.

The mentioned software project is called ChemBrain IXL, available from Neuronix Software (www.neuronix.ch (accessed on 17 February 2021), Rudolf Naef, Lupsingen, Switzerland).

## Figures and Tables

**Figure 1 molecules-26-01045-f001:**
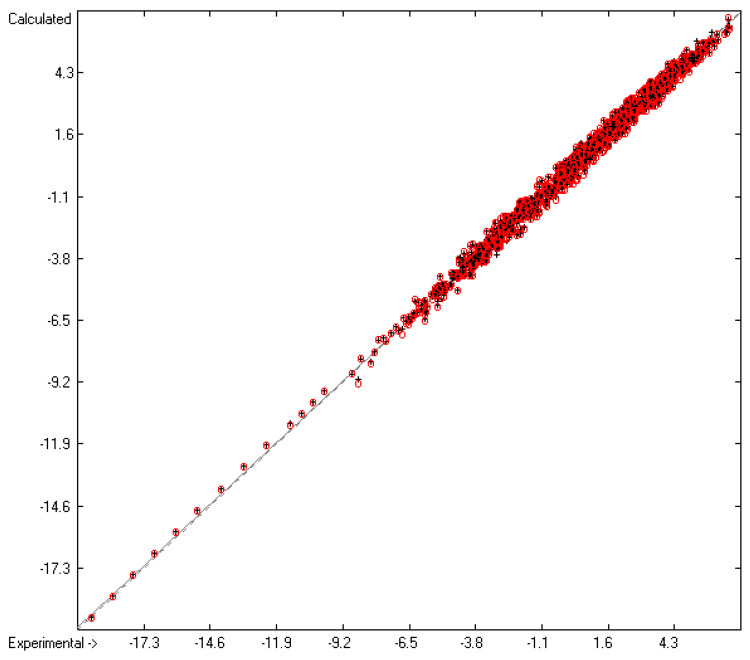
Correlation Diagram of the logVP data at 298.15 K. Cross-validation data are added as red circles. (N = 1907; R^2^ = 0.9945; Q^2^ = 0.9938; regression line: intercept = −0.0001; slope = 0.9924).

**Figure 2 molecules-26-01045-f002:**
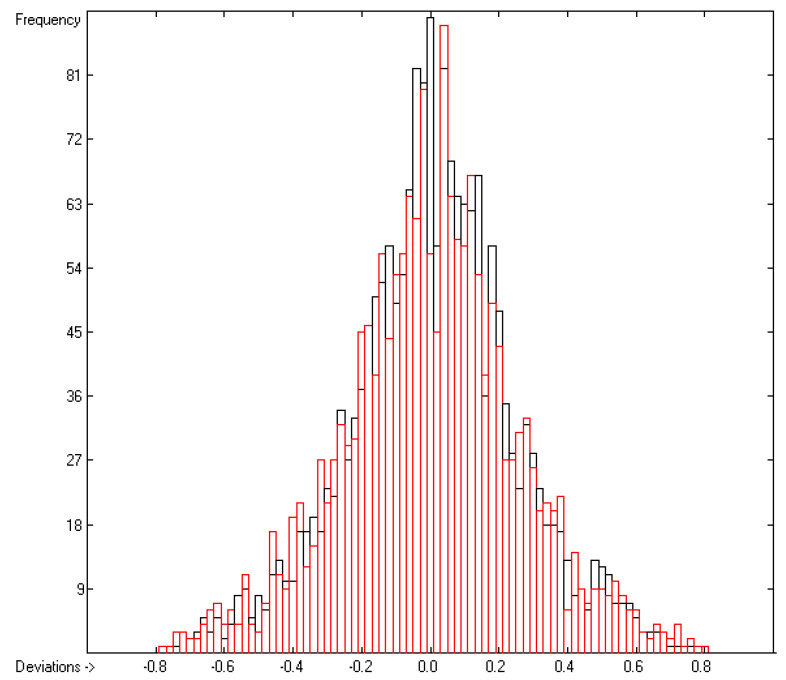
Histogram of the logVP data at 298.15 K. Cross-validation data are superpositioned as red bars. (σ = 0.24; S = 0.26; experimental values range: −19.36–+6.591).

**Figure 3 molecules-26-01045-f003:**
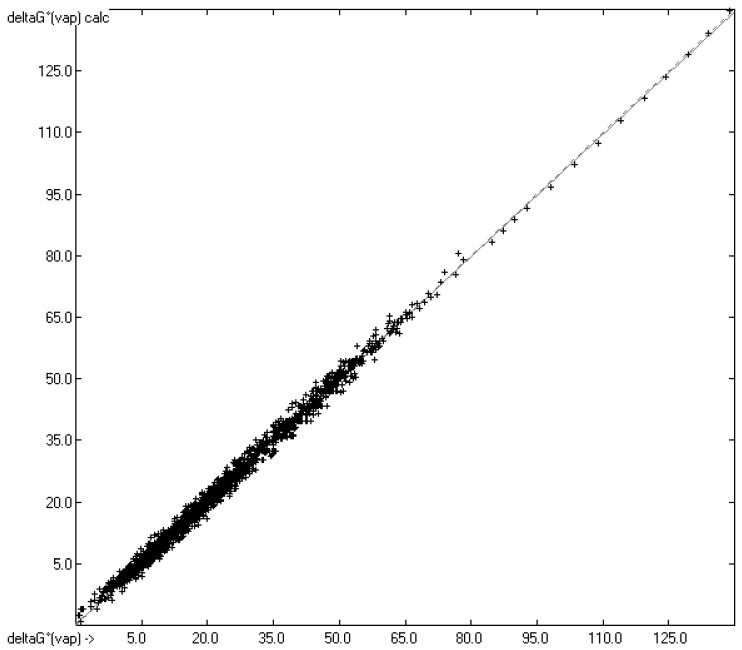
Correlation Diagram of the ΔG°_vap_ data in kJ/mol at 298.15 K. (*N* = 1907; R^2^ = 0.9945; σ = 1.38 kJ/mol; MAPD = 14.2%; regression line: intercept = 0.2167; slope = 0.9924).

**Figure 4 molecules-26-01045-f004:**
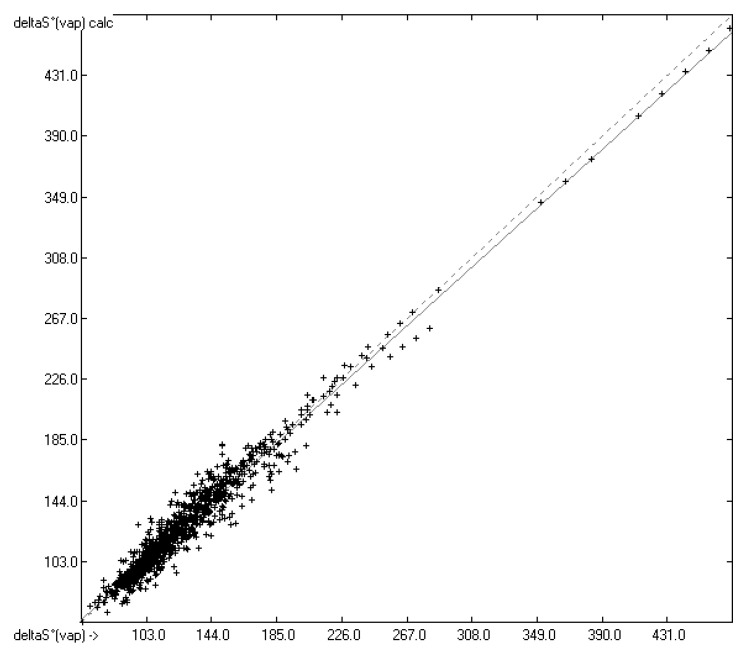
Correlation Diagram of the ΔS°_vap_ data in J/mol/ K at 298.15 K. (N = 1129; R^2^ = 0.9598; MAPD = 4.68%; regression line: intercept = 4.0448; slope = 0.9660).

**Figure 5 molecules-26-01045-f005:**
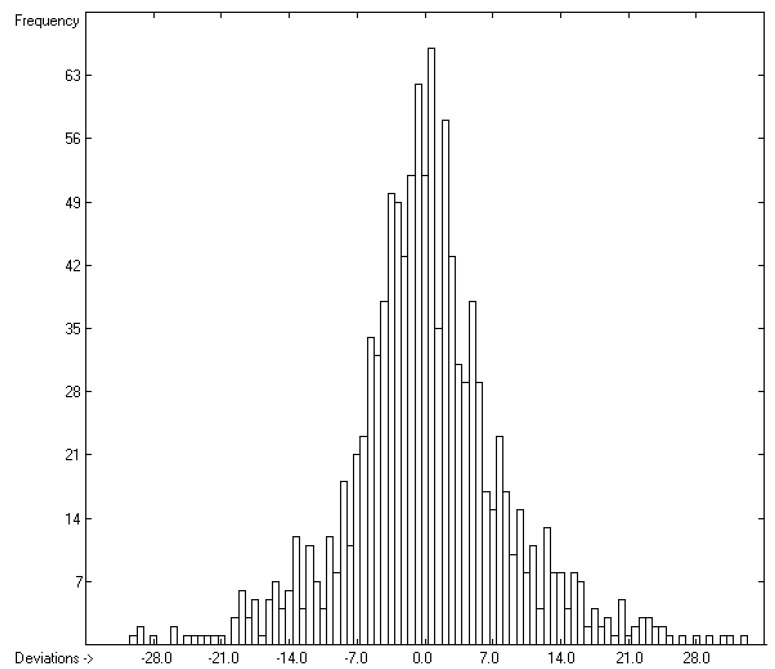
Histogram of the ΔS°_vap_ data in J/mol/ K at 298.15 K. (σ = 8.14 J/mol/K; values range: 64.26–463.16 J/mol/K).

**Table 1 molecules-26-01045-t001:** Refined atom and special groups and their meaning.

Atom Type	Neighbours	Meaning
O(prim)	HC	Primary alcohol
O(sec)	HC	Secondary alcohol
O(tert)	HC	Tertiary alcohol
(COH)n	n > 1	Molecule contains more than 1 saturated OH group
(COOH)n	n > 1	Molecule contains more than 1 carboxylic acid group
Endocyclic bonds	No of single bonds	Number of single bonds in cyclic ring
Bridgehead atoms	No of bonds	Number of bridgehead C or N (e.g., camphor, DABCO)

**Table 2 molecules-26-01045-t002:** Example calculation of the logVP of 2-Methylcyclohexanol.

Atom Type/Neighours	C sp3/H3C	C sp3/HC3	C sp3/HC2O	C sp3/H2C2	O(sec)/HC	Endocyclic Single bds	Const	Checksum
**Contribution**	0.60	−1.28	−2.65	−0.47	0.72	0.31	4.71	
**n**	1	1	1	4	1	6	1	
**n x Contrib.**	0.6	−1.28	−2.65	−1.88	0.72	1.86	4.71	**2.08**

**2.08** is the horizontal checksum.

**Table 3 molecules-26-01045-t003:** Atom groups and their contributions for logVP calculations.

Entry	Atom Type	Neighbours	Contribution	Occurrences	Molecules
1	Const		4.71	2036	2036
2	B	HN2	−1.17	6	2
3	B	BN2	−1.6	2	1
4	B	BO2	−1.71	4	2
5	B	C2N	−0.35	1	1
6	B	C2O	−0.44	1	1
7	B	C2S	−0.44	1	1
8	B	CO2	−1.56	1	1
9	B	O3	−1.57	6	6
10	B	S3	−3.17	1	1
11	C sp3	H3B	0	7	4
12	C sp3	H3C	0.6	2211	1077
13	C sp3	H3N	−1.07	113	62
14	C sp3	H3N(+)	−1.64	1	1
15	C sp3	H3O	−0.95	152	116
16	C sp3	H3S	−0.52	23	17
17	C sp3	H3P	−1.22	8	7
18	C sp3	H3Si	−0.42	87	16
19	C sp3	H2C2	−0.47	4196	831
20	C sp3	H2CN	−2.07	240	138
21	C sp3	H2CN(+)	−2.01	5	5
22	C sp3	H2CO	−1.82	460	314
23	C sp3	H2CP	−2.3	5	3
24	C sp3	H2CS	−1.6	79	54
25	C sp3	H2CF	0.39	15	15
26	C sp3	H2CCl	−0.48	59	48
27	C sp3	H2CBr	−0.76	22	20
28	C sp3	H2CJ	−1.23	11	11
29	C sp3	H2CSi	−1.58	11	6
30	C sp3	H2N2	−11.73	1	1
31	C sp3	H2NO	−3.82	2	2
32	C sp3	H2NS	−1.24	3	3
33	C sp3	H2O2	−3.81	6	6
34	C sp3	H2OF	−1.22	3	3
35	C sp3	H2OCl	−2.1	2	2
36	C sp3	H2S2	−2.59	3	3
37	C sp3	HC3	−1.28	342	231
38	C sp3	HC2N	−2.87	35	28
39	C sp3	HC2N(+)	−2.87	3	3
40	C sp3	HC2O	−2.65	115	95
41	C sp3	HC2S	−2.36	11	8
42	C sp3	HC2F	−0.6	10	9
43	C sp3	HC2Cl	−1.22	31	15
44	C sp3	HC2Br	−1.59	16	12
45	C sp3	HC2J	−1.96	1	1
46	C sp3	HCN2	−2.07	2	1
47	C sp3	HCNO	−5.99	1	1
48	C sp3	HCNS	−2.48	1	1
49	C sp3	HCO2	−3.65	7	7
50	C sp3	HCOBr	−4.78	1	1
51	C sp3	HCF2	0.37	31	27
52	C sp3	HCFCl	−0.01	7	7
53	C sp3	HCCl2	−0.94	12	11
54	C sp3	HCClBr	−0.76	1	1
55	C sp3	HCBr2	−1.93	3	2
56	C sp3	HOF2	−1.09	6	6
57	C sp3	C4	−2.19	98	87
58	C sp3	C3N	−3.6	11	11
59	C sp3	C3N(+)	−3.57	2	2
60	C sp3	C3O	−3.46	36	35
61	C sp3	C3S	−3.21	6	6
62	C sp3	C3Si	−3.37	3	2
63	C sp3	C3Cl	−2.84	6	3
64	C sp3	C3Br	−2.2	2	2
65	C sp3	C3F	−1.39	13	10
66	C sp3	C2O2	−5.46	4	2
67	C sp3	C2OF	−2.8	5	5
68	C sp3	C2F2	−0.37	184	71
69	C sp3	C2FCl	−0.8	1	1
70	C sp3	C2Cl2	0	3	3
71	C sp3	CNF2	−2.03	12	5
72	C sp3	CNF2(+)	−0.37	1	1
73	C sp3	CNCl2	−0.4	1	1
74	C sp3	COF2	−1.69	49	39
75	C sp3	CSF2	−1.15	24	12
76	C sp3	CF3	0.67	152	107
77	C sp3	CF2Cl	0.3	8	7
78	C sp3	CF2Br	−0.07	5	4
79	C sp3	CFCl2	−0.37	5	4
80	C sp3	CFClBr	−0.73	1	1
81	C sp3	CCl3	−0.98	15	14
82	C sp3	CCl2Br	0	1	1
83	C sp3	NF3	−1.09	5	3
84	C sp3	OF3	−0.36	13	10
85	C sp3	O2F2	−2.67	1	1
86	C sp3	S2F2	−1.83	4	2
87	C sp3	SF3	−0.01	10	7
88	C sp3	SCl3	−7.92	1	1
89	C sp3	PF3	−0.08	20	8
90	C sp2	H2=C	0.67	127	113
91	C sp2	HC=C	−0.38	272	175
92	C sp2	HC=N	−1.49	7	7
93	C sp2	HC=O	−0.47	27	27
94	C sp2	H=CN	−1.84	19	12
95	C sp2	H=CO	−0.79	5	5
96	C sp2	H=CS	−0.79	8	6
97	C sp2	H=CP	−1.03	3	1
98	C sp2	H=CF	0.68	3	3
99	C sp2	H=CCl	−0.15	13	11
100	C sp2	H=CBr	−0.56	5	3
101	C sp2	H=CJ	−1.2	2	1
102	C sp2	HN=N	−1.89	11	9
103	C sp2	HN=O	−2.47	9	8
104	C sp2	HO=O	−1.25	8	8
105	C sp2	C2=C	−1.25	79	67
106	C sp2	C2=N	−3.09	2	2
107	C sp2	C=CN	−2.26	2	2
108	C sp2	C2=O	−1.27	56	53
109	C sp2	C=CO	−1.5	6	6
110	C sp2	C=CP	−3.09	1	1
111	C sp2	C=CS	−1.78	6	5
112	C sp2	C=CF	−0.25	3	3
113	C sp2	C=CCl	−1.24	18	13
114	C sp2	CN=N	−4.13	2	2
115	C sp2	CN=O	−3.17	35	32
116	C sp2	C=NS	−1.47	2	1
117	C sp2	CO=O	−2.33	222	184
118	C sp2	C=OCl	−0.54	4	4
119	C sp2	C=OBr	−1.1	1	1
120	C sp2	C=OJ	−1.67	1	1
121	C sp2	=CF2	0.95	7	6
122	C sp2	=CFCl	0.14	1	1
123	C sp2	=CFBr	−0.25	1	1
124	C sp2	=CCl2	−0.53	10	8
125	C sp2	=CBr2	0.65	1	1
126	C sp2	N2=N	−4.7	1	1
127	C sp2	N2=O	−5.19	5	5
128	C sp2	N=NS	−1.58	1	1
129	C sp2	N2=S	0.14	2	1
130	C sp2	NO=O	−4.55	15	13
131	C sp2	N=OS	−0.64	7	7
132	C sp2	=NOS	−0.26	1	1
133	C sp2	NS=S	1.24	1	1
134	C sp2	O2=O	−3.59	4	4
135	C aromatic	H:C2	−0.2	3662	751
136	C aromatic	H:C:N	−0.41	34	21
137	C aromatic	H:N2	0.48	2	2
138	C aromatic	:C3	−1.06	260	85
139	C aromatic	C:C2	−1.06	929	508
140	C aromatic	C:C:N	−1.21	15	13
141	C aromatic	:C2N	−2.3	40	38
142	C aromatic	:C2N(+)	−2.59	33	29
143	C aromatic	:C2:N	−1.42	4	3
144	C aromatic	:C2O	−2	381	195
145	C aromatic	:C2P	−4.07	1	1
146	C aromatic	:C2S	−1.69	8	6
147	C aromatic	:C2F	−0.1	63	26
148	C aromatic	:C2Cl	−0.84	1630	386
149	C aromatic	:C2Br	−1.13	166	58
150	C aromatic	:C2J	−1.57	10	9
151	C aromatic	:C2Si	0.89	1	1
152	C aromatic	C:N2	−1.39	2	2
153	C aromatic	:C:NO	−1.85	6	6
154	C aromatic	:C:NCl	−1.33	5	5
155	C aromatic	N:N2	−2.72	17	10
156	C aromatic	:N2O	−0.96	2	2
157	C aromatic	:N2S	1.89	3	3
158	C aromatic	:N2Cl	−1.38	3	3
159	C sp	H#C	0.81	14	13
160	C sp	C#C	−0.49	22	17
161	C sp	=C2	−0.51	3	3
162	C sp	C#N	−0.61	34	27
163	C sp	=N=O	0.75	3	3
164	C sp	=N=S	1.19	1	1
165	N sp3	HB2	0.45	3	2
166	N sp3	H2C	1.45	62	47
167	N sp3	H2C(pi)	0.15	18	18
168	N sp3	H2N	−0.52	3	3
169	N sp3	HC2	2.36	26	26
170	N sp3	HC2(pi)	0.56	35	26
171	N sp3	HC2(2pi)	0.44	14	10
172	N sp3	HCN	0.7	3	2
173	N sp3	HCN(pi)	−0.36	1	1
174	N sp3	HCN(2pi)	0.38	1	1
175	N sp3	HCP(pi)	−4.25	1	1
176	N sp3	HCS(pi)	5.56	1	1
177	N sp3	B2C	1.1	3	2
178	N sp3	BC2	2.13	5	2
179	N sp3	C3	3.52	49	45
180	N sp3	C3(pi)	2.95	27	26
181	N sp3	C3(2pi)	3.52	11	11
182	N sp3	C3(3pi)	3.4	3	3
183	N sp3	C2N(pi)	0.11	4	4
184	N sp3	C2N(2pi)	3.37	8	8
185	N sp3	C2N(3pi)	2.89	1	1
186	N sp3	C2O	3.47	1	1
187	N sp3	C2S	2.57	3	3
188	N sp3	C2S(pi)	3.96	3	2
189	N sp3	C2S(2pi)	7.1	1	1
190	N sp3	C2P	2.07	7	4
191	N sp3	C2F(pi)	4.38	1	1
192	N sp3	CF2	0.61	1	1
193	N sp3	CSi2	1.18	2	2
194	N sp3	SF2	0.07	1	1
195	N sp2	C=C	0.39	16	15
196	N sp2	C=N	−3.19	1	1
197	N sp2	C=N(+)	0.96	7	7
198	N sp2	=CN	−0.04	10	9
199	N sp2	=CO	0.68	4	3
200	N sp2	=CS	−0.39	1	1
201	N sp2	N=N	0	1	1
202	N sp2	N=O	0	4	4
203	N sp2	=NP(+)	−0.39	1	1
204	N sp2	O=O	1.58	6	6
205	N aromatic	:C2	−0.06	61	39
206	N(+) sp2	CO=O(−)	0.34	45	41
207	N(+) sp2	O2=O(−)	0.54	50	26
208	N(+) sp	=N2(−)	0	8	8
209	O(prim)	HC	0.44	95	78
210	O(sec)	HC	0.72	48	47
211	O(tert)	HC	0.74	11	11
212	O	HC(pi)	0.04	102	90
213	O	HN(pi)	−1.29	1	1
214	O	HO	−1.16	4	3
215	O	BC	1.39	26	8
216	O	BP	0.16	3	2
217	O	C2	2.38	150	132
218	O	C2(pi)	2.3	228	191
219	O	C2(2pi)	1.49	151	130
220	O	CN	0	1	1
221	O	CN(pi)	0	6	6
222	O	CN(2pi)	0.26	3	2
223	O	CN(+)(pi)	0	50	26
224	O	CO	1.03	8	3
225	O	CO(pi)	1.59	3	2
226	O	CS	1.25	6	4
227	O	CS(pi)	1.44	2	2
228	O	CP	0.06	95	44
229	O	CP(pi)	−0.29	14	12
230	O	CSi	0.65	7	2
231	O	OS	−0.67	3	2
232	O	S2	−1.14	5	3
233	O	Si2	−0.3	22	7
234	P3	C3	0	1	1
235	P3	HC2	2.57	1	1
236	P3	C2N	1.59	2	2
237	P3	C2O	0	3	2
238	P3	C2S	−0.09	5	4
239	P3	CN2	−0.35	1	1
240	P3	CS2	−0.94	1	1
241	P4	HO2=O	−0.55	1	1
242	P4	C3=S	0.19	1	1
243	P4	CO2=O	0.62	4	4
244	P4	CO2=S	3.03	1	1
245	P4	CO=OS	0.38	2	2
246	P4	COS=S	−0.5	1	1
247	P4	N3=O	−0.83	1	1
248	P4	NO=OS	−0.06	1	1
249	P4	N=OF2	0	1	1
250	P4	O3=O	0.23	9	9
251	P4	O3=S	0.22	13	13
252	P4	O2=OS	−0.36	1	1
253	P4	O=OS2	−1.76	1	1
254	P4	O2S=S	−0.58	12	11
255	S2	HC	0.83	29	23
256	S2	HC(pi)	0.28	1	1
257	S2	HS	−0.26	2	1
258	S2	HP	0.06	1	1
259	S2	BC	0.52	4	2
260	S2	C2	1.07	30	28
261	S2	C2(pi)	−1.97	14	13
262	S2	C2(2pi)	1.52	9	9
263	S2	CN	0	1	1
264	S2	CN(2pi)	−2.3	1	1
265	S2	CS	0.05	8	4
266	S2	CP	−0.07	22	19
267	S2	CP(pi)	0	1	1
268	S2	N2	−1.45	2	2
269	S2	NCl	−0.43	1	1
270	S2	P2	−0.7	1	1
271	S2	Si2	0.33	3	3
272	S4	C2=O	−0.96	4	4
273	S4	C2=O2	1.6	2	2
274	S4	C2O2	−2.15	1	1
275	S4	C2F2	0.41	5	5
276	S4	CO=O2	2.16	1	1
277	S4	CN=O2	−2.13	1	1
278	S4	NO=O2	−2.51	1	1
279	S4	N=O2Cl	0	1	1
280	S4	O2=O	−0.56	1	1
281	S4	O2=O2	−0.94	1	1
282	S4	O=O2F	0.12	4	4
283	S6	C2F4	0.72	5	3
284	S6	O2F4	−0.78	1	1
285	S6	OF5	1.08	7	5
286	Si	H3C	1.72	4	4
287	Si	H3N	0	4	2
288	Si	H3S	−0.3	2	1
289	Si	H3Si	−0.53	2	1
290	Si	H2C2	1.78	2	2
291	Si	H2Si2	0	2	1
292	Si	HC2O	0.78	2	1
293	Si	HC2S	0.11	2	1
294	Si	HC2J	0.23	1	1
295	Si	HCCl2	0.47	1	1
296	Si	HO3	0.21	1	1
297	Si	C4	1.97	2	2
298	Si	C3O	1.08	6	3
299	Si	C3S	0.19	2	1
300	Si	C3Cl	1.05	1	1
301	Si	C3Si	−0.79	2	1
302	Si	C2O2	−0.18	18	5
303	Si	C2F2	1.69	1	1
304	Si	C2Cl2	0.41	1	1
305	Si	CF3	0	1	1
306	Si	CCl3	0.06	1	1
307	Si	O4	−0.16	1	1
308	(COH)n	n>1	−0.74	23	22
309	(COOH)n	n>1	−1.73	12	12
310	Endocyclic bonds	No of single bds	0.31	1072	193
311	Bridgehead atoms	No of atoms	0.23	80	27
312	Angle60		0.19	42	14
313	Angle90		0.17	72	21
314	Angle102		0.11	323	110
**A**	**Based on**	**Valid groups**	**171**		**2036**
**B**	**Goodness of fit**	**R^2^**	**0.9946**		**1908**
**C**	**Deviation**	**Average**	**0.18**		**1908**
**D**	**Deviation**	**Standard**	**0.24**		**1908**
**E**	**K-fold cv**	**K**	**10**		**1842**
**F**	**Goodness of fit**	**Q^2^**	**0.9938**		**1842**
**G**	**Deviation**	**Average (cv)**	**0.2**		**1842**
**H**	**Deviation**	**Standard (cv)**	**0.26**		**1842**

Lines A to H are the statistics data of the table.

## Data Availability

Not applicable.

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
