# Peer review of "Calculation of the Vapour Pressure of Organic Molecules by Means of a Group-Additivity Method and Their Resultant Gibbs Free Energy and Entropy of Vaporization at 298.15 K"

_molecules, 2021, doi:10.3390/molecules26041045_

Round 1

Reviewer 1 Report

- Intro -- Authors should briefly describe the method outlined in ref 17

- For the COOH group clarify if a hydro peroxide or acid group,

- Relative to the initial part of the article (before further added descriptions) if the COOH group is a carboxylic acid one might ask at this point in the manuscript - how can this carbon have more than one neighbor or clarify that this is a single group.

- p 4  Do the cyclohexanol as a table could be 1 or 2 or 3 columns to conserve space (lines). Better yet do the list of groups and the values reported just below in the table to show more clearly.

One might consider using the letter N or n for number e.g. N (i) ?

Author Response

Please read my answers in the attached PDF file.

Reviewer 2 Report

This work extends a well-established group additivity framework for chemical property prediction to yet another property. The focus is on the vapor pressure of organic molecules. There have been many efforts to model vapor pressures before for different molecular classes, as summarized in the introduction. It is good extension of the property database and the group additive framework; however, the work is lacking some novelty. The manuscript is well written and the results are presented in a detailed manner. Below are some comments/questions:

Method

  1. Can the authors give some more information on the database structure, which software/language/data format is used? How are 3D geometries optimized? Are unique identifiers stored? Is the complete database available to the public?
  2. Are ionic species considered in this method? Are there additional group contributions for ions?
  3. The authors discuss that primary/secondary/tertiary alcohols need to be distinguished and the need for secondary contributions if more alcohol groups are present. Is there an influence of how far these group are apart in the considered molecule? Is this an effect that has been seen for amines as well? If not, do the authors think this effect is just not present in amines or is there not sufficient data to see the effect?
  4. Please mention the meaning of the Anglex contribution in the methodology.

Results

  1. Do the authors notice a difference between the vapor pressures of compounds that are stable in the liquid phase or in the solid phase at 298.15K? Does the method work well for both phases? And how is the data biased to either one of those?
  2. The main question is how the performance of the group additive framework compares to other published methods. For example, to the EVAPORATION mode by Compernolle et al. Some errors from published methods are summarized in the introduction, but it would be helpful to have an overview and short discussion.

Author Response

(The authors gave the same response as above.)

Reviewer 3 Report

The manuscript deals with the calculation of vapor pressure of organic compounds by a general computer algorithm based on the group additivity method. This is part of a set of related papers. In the previous papers the the same basic computer algorithm based on the atom-group additivity method was applied to predict other 16 molecular descriptors. In this manuscript, certain previous atom groups were replaced by more detailed ones and given special groups were added or omitted to improve the vapor pressure prediction capability. The manuscript is recommended to be accepted to publication. However a revision for typos is needed, for example: in page 3 - "for compounds with with n>1"; in page 6 - Pascal should be used instead of pascal, etc. Also in table 2,  3rd column (neighbours), what does J mean (entries 28, 45, 101, 120, 150 and 294)?   

Author Response

(The authors gave the same response as above.)

Reviewer 4 Report

This manuscript presents an extension of a group-additivity approach to estimate the vapour pressure and, as a consequence, the Gibbs free energy of vaporization of a large data base of molecules. The results computed with the contribution method agree well with the experimental data.

Overall, this study addresses an interesting research topic. The paper is well-written, and the approach is mostly well-described. It can be accepted for publication, provided the following minor issues are addressed satisfactorily.

  • The difference between experimental and calculated properties (e.g. vapor pressure, free energy, entropy) is reported in the supporting information in terms of absolute deviation. I think it would be more appropriate to show a relative deviation (for example in %).
  • Related to the previous point, deviations between calculated and experimental values exceeding 3 times the standard error were excluded from further calculations. Is there a reason why this criterion was chosen instead of a threshold on a relative deviation?
  • The successful prediction of the vapor pressure allowed the extraction of both Gibbs free energy and standard entropy of vaporization. From Fig. 3 and 4, the prediction of the Gibbs free energy of vaporization seems more more accurate than the corresponding one for the entropy. However, a lower MAPD is reported in this case. What is the reason for this discrepancy?

Author Response

(The authors gave the same response as above.)
